# Herpes Zoster in an Immunocompetent Child without a History of Varicella

**Bing-Shiau Shang** [1,2], **Cheng-Jui Jamie Hung** [2] and **Ko-Huang Lue** [1,2,*]

1    Department of Pediatrics, Chung Shan Medical University Hospital, Taichung City 402, Taiwan; j6ji3@hotmail.com
2    Department of Medicine, Chung Shan Medical University, Taichung City 402, Taiwan; chengjui.jamie.hung@gmail.com
*    Correspondence: cshy095@csh.org.tw

**Abstract:** Herpes zoster is a relatively rare infectious disease in the pediatric population, as compared with adults, which is due to the reactivation of latent Varicella−Zoster virus. We report a 7-year-old child without any history of varicella, who first experienced skin pain and later presented skin lesions in dermatomal distribution. Finally, the patient was diagnosed with herpes zoster. We aim to emphasize that herpes zoster could occur in immunocompetent children and may be due to the reactivation of the vaccine strain or previous subclinical infection.

**Keywords:** varicella; herpes zoster; vaccine; pediatric; immunocompetent





## 1. Introduction

Herpes zoster (HZ), also known as shingles, is a rather uncommon disease in pediatrics which is caused by the reactivation of the latent Varicella−Zoster virus (VZV) [1]. Despite the fact that most patients with HZ have a medical history of varicella, or chickenpox (the manifestation of the primary infection of VZV), there is a group of pediatric populations that has HZ without any record of varicella (infection). Meanwhile, as live attenuated varicella vaccine is routinely administered in many countries worldwide, there are reports stating that HZ has appeared in immunocompetent children after varicella vaccination [2]. We herein report a case of a child of HZ, who received live attenuated varicella vaccine at 12 months old and had no history of varicella.

## 2. Case Report

This was a 7-year-7-month-old immunocompetent boy with no past history of varicella, or chickenpox, or any other systemic diseases. He was born at a gestational age of 36 weeks via Caesarean section (C-section), due to previous C-section with a maternal history of Gravida 3 Para 3. He received the varicella vaccine, when he was 1 year old which was scheduled in the routine childhood vaccination program. The patient was brought to our emergency department on October 4, 2020 due to multiple vesicles and erythematous rashes over his right buttock and right lower leg after a three-day history of severe burning pain and tenderness, especially during the night, in his right leg. He was hemodynamically stable and had no fever or any neurologic abnormality. Based on the appearance of the patient's cutaneous lesions, the patient was admitted to our pediatric ward on the same day.

Upon admission, the physical examination revealed numerous fine vesicles on an erythematous base, which distributed approximately around L3 and L4 dermatomes (Figure 1a,b). We noted an antalgic gait, and the patient experienced extensive pain in his right leg. Blood laboratory examinations indicated no leukopenia, leukocytosis or an elevated C-reactive protein (CRP) level. Serologic screening for VZV antibodies (Ab) was also arranged which showed a positive result of VZV IgG Ab and a gray zone result of VZV IgM Ab.

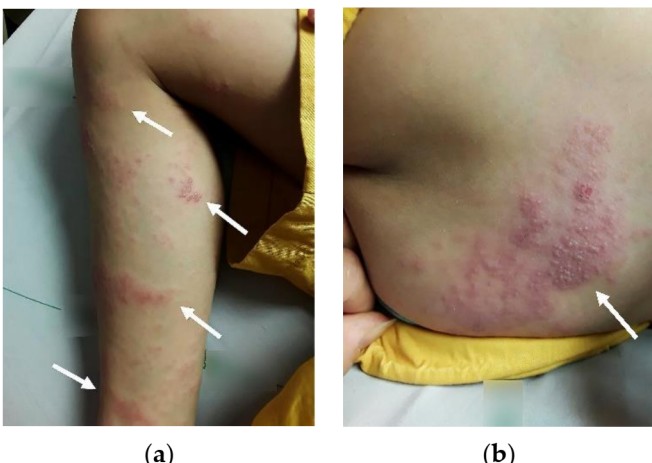

(**a**) (**b**)

**Figure 1.** (**a**,**b**) Multiple fine vesicles with erythematous bases distributed around right L3 and L4 dermatomes.

Intravenous infusion of acyclovir 10 mg/kg/dose every 8 h and neurologic pain control with oral gabapentin 10 mg/kg/day every 8 h and ibuprofen 15 mg/kg/day every 6 h were prescribed on the first day of the admission. Though more clusters of vesicles appeared on the second and the third admission day, we kept the antiviral treatment throughout the entire clinical course. However, since the patient still complained about persistent burning pain, tightness in the anterior part of his right thigh, we replaced gabapentin with oral pregabalin. The patient also complained of some itching sensation over the vesicular lesions. After the fourth day of admission, the vesicular lesions gradually became dryer, darker and more crusted; no new skin lesions were observed. On 15 October, the twelfth admission day, the patient was discharged with the final diagnosis of herpes zoster. The patient visited our out-patient department for follow-up three weeks after he was discharged, and he no longer felt any pain in his right leg so that postherpetic neuralgia (PHN) was ruled out.

## 3. Discussion

Since the introduction of the live attenuated varicella vaccine, also known as the Oka vaccine, the incidence of herpes zoster (HZ) in children has declined from 20–63/100,000 to 14/100,000 person-years [3,4]. It is known that HZ in children is rare, in which most patients diagnosed with HZ are immunocompromised or under pharmaceutical treatment with immunosuppressive drugs [5]. On the other hand, while the reactivation of latent VZV, causing HZ, is more common in adults than in pediatric patients, their respective symptoms have slight differences. Whereas pain in dermatomal distribution is mostly complained about by adults with HZ, itching, followed by pain, fever, and weakness, is the most common symptom in children with HZ [6].

Clinical manifestations of HZ in immunocompetent children have been reported. We reviewed 27 previous reports in literature, which comprised 39 immunocompetent children with HZ, all of which were related to a history of either varicella or varicella vaccination (Table 1) [7–33]. In these 39 reported cases, there were 23 male patients and 15 female patients, while one case report did not reveal the sex of the patient. The age of acquiring HZ ranged from 1 year 3 months to 16 years old. Whereas six patients had no history of varicella vaccination, the interval between varicella vaccination and the presentation of HZ varied from 56 days to nearly 9 years. The affected cutaneous region with HZ presentation included various dermatomes, including cervical, thoracic, lumbar, sacral, and trigeminal involvement. There was no distinct connection between the vaccine injection site and the cutaneous lesions of HZ. However, a retrospective study, conducted by Aktas et al., demonstrated that HZ in children younger than 10 years old was more frequently involved with cervical, trigeminal, and sacral dermatomes, while HZ in older children showed a tendency to present thoracic or lumbar involvement [6].

**Table 1.** Previous Literature of Herpes Zoster in Pediatric Patients.

| Reference | Age | Sex | Age at Vaccination | Varicella History | Interval between Vaccination and HZ | Dermatomes/Regions with Lesions | VZV Strain | IgM | IgG |
|---|---|---|---|---|---|---|---|---|---|
| Na, G. Y. et al. (1997) | 4 y 10 m | M | 2 y | No | 2 y 10 m | R't S3 | Wild type | - | + |
| | 3 y 1 m | F | 1 y 5 m | No | 1 y 8 m | L't S3–4 | Wild type | - | + |
| | 5 y 5 m | M | 3 y 5 m | No | 2 y | R't T2 | Wild type | - | ND |
| | 2 y 6 m | F | 1 y | No | 1 y 6 m | L't T10 | ND | - | + |
| | 3 y 9 m | F | 2 y 5 m | No | 1 y 3 m | R't L3–4 | ND | - | + |
| | 4 y 2 m | M | 2 y 4 m | No | 1 y 10 m | R't S1–2 | ND | - | + |
| Liang, M. G. et al. (1998) | 1 y 7 m | F | 1 y 3 m | Yes (7 m) | 4 m | L't C6–7 | Vaccine | ND | ND |
| Kohl, S. et al. (1999) | 6 y | M | 6 y | No | 12 d | L't T2 | Wild type | ND | ND |
| Uebe, B. et al. (2002) | 2 y 3 m | F | 11 m | No; contact history with her sister with varicella | 1 y 4 m | R't C6–C8 | Vaccine | - | + |
| Feder, H. M. et al. (2004) | 6 y | F | N/A | Yes (3 m) | N/A | R't V2–3 | ND | ND | ND |
| | 3 y | F | N/A | Yes (6 m) | N/A | L't L1–2 | ND | ND | ND |
| | 8 y | F | N/A | Yes (7 y) | N/A | R't T6 | ND | ND | ND |
| | 3 y | M | 1 y | Yes (6 m) | 2 y | L't V1 | ND | ND | ND |
| | 5y | F | 1 y 3 m | N/A | 3 y 9 m | L't L2 | Wild type | ND | ND |
| Ota, K. et al. (2008) | 2 y 4 m | M | 1 y 1 m | N/A | 1 y 3 m | L't chest and upper limb | Vaccine | ND | + |
| Levin, M. J. et al. (2008) | 8 y | M | 1 y | No | 7 y | R't shoulder and meningitis | Vaccine (skin and CSF) | + | + |
| Iyer, S. et al. (2009) | 9 y | M | 1 y | No | 8 y | L't C5–6 & meningitis | Vaccine (Skin and CSF) | ND | ND |
| Chouliaras G. et al. (2010) | 3 y6 m | F | 1 y 8 m | No; contact history at 1 y 3 m | 1 y 10 m | R't V1 and encephalitis | Vaccine (CSF) | ND | ND |
| Han, J. Y. et al. (2011) | 7 y | M | 1 y | No | 6 y | R't arm and meningitis | Vaccine (Skin and CSF) | ND | ND |
| Iwasaki, S. et al. (2016) | 2 y | F | 1 y 5 m | No | 7 m | L't V1–2 | Vaccine | + | + |
| Dreyer, S. et al. (2017) | 3 y | M | 1 y | N/A | 2 y | R't L2 | Vaccine | ND | ND |
| | 2 y | F | 1 y | No | 1 y | L't L4 | Vaccine | ND | ND |
| Moodley, A. et al. (2018) | 3 y 3 m | M | 1 y 8 m | No | 1 y 7 m | L't L4-S1 | Wild type variant of vaccine | ND | ND |
| | 1 y 8 m | M | 1y 1 m | No | 7 m | R't L3 | ND | + | + |
| | 3 y 6 m | M | 1 y | No | 2 y 6 m | R't thigh | ND | N/A | N/A |
| Pelekouda, E. et al. (2019) | 4 y | F | 1 y 3 m | No | 2 y 9 m | R't C4–5 & T1 | ND | - | ND |
| Yasuda, R. et al. (2019) | 11 y | F | N/A | Yes (2 y) | N/A | L't chest and meningitis | ND | - | + |
| Harrington, W. E. et al. (2019) | 14 y | M | 1 y and 4 y | No | ? | L't L1–2 and meningitis | Vaccine (Skin and CSF) | ND | ND |

The references were listed in chronological order. Abbreviations: y = year, m = month, d = day; F = female, M = male; ND = not done in the reported case; N/A = information was not available. The question mark (?) indicates the reported case received more than one dose of varicella vaccine so that the interval between vaccination and HZ was unclear. The symbol "+" indicates positive finding, and the symbol "−" indicates negative finding.

HZ is basically diagnosed alongside the clinical presentation and is likely to be derived from the reactivation of either wild type or vaccine strain in immunocompetent children. Some previous cases also performed virology screening that wild-type VZV was identified in six patients and vaccine strain was found in 11 patients, which indicates that varicella vaccine has the potential for causing not only latent infection, but also reactivation. Meanwhile, it was reported that five patients were infected with wild-type VZV without previous exposure to varicella or HZ or any other contact history. Thus, HZ in children may also arise from subclinical infection of VZV [7,9,11]. Additionally, despite clustered vesicular lesions in dermatomal distribution being the most common symptom of HZ in children, previous reports also include cases of central nervous system (CNS) infection, in which either wild type or vaccine strain was found in the virology screening for the cerebrospinal fluid (CSF) and further led to meningitis or encephalitis [13,16,18,19,34]. It is suggested that both wild type and vaccine strain VZV reactivation, can result in severe complications, especially CNS involvement.

Laboratory examination has not been identified as a necessary process for the diagnosis or treatment of HZ [1]. Therefore, in the reviewed articles, VZV-specific IgM tests were performed in 14 cases and positive results of three cases; while VZV-specific IgG test were performed in 13 cases and positive results of all cases. Specific IgM was positive in three cases, two of which were caused by the vaccine strain [7,10,13,22,27,29,31,32]. In fact, Kangro et al. and Min et al. suggested HZ would show an IgM-positive result [35,36]. Min et al. also found that the IgM level usually peaked around the 6th to 10th day after the cutaneous lesions developed and could persist up to 10 weeks [36]. Nevertheless, their studies were not designed specifically for the pediatric population, whose immune response to viral infection is not completely identical to adults. There has been no previous literature discussing the positive duration of IgM in children with HZ particularly either.

Our patient was a 7-year-old boy with HZ involvement around right L3 and L4 dermatomes, which does not correspond with the previous finding that patients under 10 years old are more likely to present HZ in cervical, sacral, and trigeminal dermatomes [6]. The serologic test of our patient showed a positive-IgG result and a gray zone (1.01) result of IgM (<0.8 as negative, >1.1 as positive). We supposed the slight increase of IgM level was due to the unreliability of the currently available IgM detection technique, or the early examination of IgM, level of which usually rises on the 6th to 10th day after the appearance of vesicular lesions [36]. As he received a dose of varicella vaccine at 1 year old and denied a history of varicella, it was suspected that the HZ was caused by subclinical infection or vaccine strain VZV. Yet due to personal reasons, the patient's family opposed the arrangement of confirmatory laboratory examination to identify the strain of the VZV infection, which consequently becomes a limitation of the report.

Furthermore, certain gene mutations may be susceptible to specific pathogen infections or activations, including VZV. Korholz et al. had reported a dominant inherited IFNγR1 deficiency in a family [37]. The affected patient and one of his sons had recurrent and severe infections (pneumonia, meningoencephalitis, shingles) of VZV, HSV; while another affected son remained clinically healthy. Droman et al. demonstrated that a patient with IFNγR1 deficiency had over 70% of mycobacterial infection from environmental or BCG-vaccine in both dominant and recessive type, with 100% of BCG vaccine-related mycobacteria disease in recessive inherited type [38]. Our patient denied having any complications after BCG vaccination at 5 months old (routine vaccination program in Taiwan). However, in countries without a routine BCG vaccination, clinics should be alert to certain immunodeficiency even if the patient was previously healthy.

As both wild type and vaccine strain can contribute to the reactivation of latent VZV, HZ should be considered when a pediatric patient presents vesicular lesions in dermatomal distribution, even without a history of varicella or any contact history. Moreover, since HZ can cause various complications, including CNS infection, physicians should keep it in mind when approaching patients with encephalitis or meningitis. Future studies regarding the positive duration of VZV IgM and IgG levels with a focus on the pediatric

population in particular may also be helpful to ascertain the effectiveness and reliability of the serologic test.

**Author Contributions:** Conception and design, K.-H.L.; drafting the article, B.-S.S. and C.-J.J.H.; revising the article critically for important intellectual content, B.-S.S.; final approval of the version to be published, K.-H.L. All authors have read and agreed to the published version of the manuscript.

**Funding:** This research did not receive any specific grant from funding agencies in the public, commercial or not-for-profit sectors.

**Institutional Review Board Statement:** Not applicable.

**Informed Consent Statement:** Informed consent was obtained from all subjects involved in the study.

**Data Availability Statement:** Not applicable.

**Acknowledgments:** We would like to thank the patient and the family for their active participation in publishing this case report. All figures were taken or recorded with the patient's agreement. Consent of the patient was also obtained for publication of this case report from the family.

**Conflicts of Interest:** The authors declare no conflict of interest.

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
