# Peer review of "Herpes Zoster in an Immunocompetent Child without a History of Varicella"

_pediatrrep, doi:10.3390/pediatric13020022_

Round 1
Reviewer 1 Report
The Case report of Bing-Shiau Shang et al. describes an interesting case of HZ in a Immunocompetent child. The paper is well written and easily understandable, but would benefit from some grammatical corrections as well as additional information.
Abstract:
L9:compared with adults, which is due.. (comma missing here)
1.Introduction:
Is very short compared to the discussion.
L18-20: Despite the fact that most patients with HZ have a medical history of varicella, or chickenpox, the manifestation of the primary infection of VZV, there is a group of pediatric populations that has HZ without any record of varicella
sentence structure is confusing, you may suggest to change it to:
Despite the fact that most patients with HZ have a medical history of varicella, or chickenpox (the manifestation of the primary infection of VZV), there is a group of pediatric populations that has HZ without any record of varicella (infection)
L23/24:
We herein report a case of a child of HZ, (comma was missing here) who received live attenuated varicella vaccine at 12 24 months old and had no history of varicella.
2. Case report:
L29: weeks via Caesarean section (C-section), (comma missing) due to previous C-section
L30: vaccine, (comma missing) when he was 1 year old which was
L39: erythematous base, (comma missing) which distributed approximately around L3 and L4 dermatomes
L40: Antalgic gaits were also noted as the patient experienced extensive pain in his right leg.
you may change the sentence here, suggestions:
We noted an antalgic gait,....
L42: VZV antibody (Ab)
plura needed here, please change to antibodies
L46: oral gabapentin and ibuprofen: what was the exact dose?
L50 : Also, there was some itch-50 ing sensation over the vesicular lesions.
Please replace the word also,...
Discussion:
In 3 % of children developing HZ a malignancy as underlying disease is discovered (especially leukemia). This might be worthwhile mentioning in the discussion, is research done in the case study?
What research was done to detect concurrent immunsuppression?
There have been several case studies who reported post covid HZ and even HZ as indicator of latent covid 19 infection (Elsaie et al Dermatologic therapy/vol33/issu4). Is a covid test performed?
L59: in children had declined from (change to has declined)
L60-61: It is known that HZ in children is rare that most pediatric patients with HZ are immunocompromised or under pharmaceutical treatment of immunosuppressive drugs
comma missing here and sentence structure is confusing, please change
L63: their respective symptoms have slight differences.
please change to differ slightly
L68: previous literatures, (comma missing) which comprise 39 immunocompetent children
L81: HZ is basically diagnosed with the clinical presentation
you may change this to: diagnosed alongside to the clinical presentation
L92-94: It is suggested, (comma missing) that both wild-type and vaccine strain VZV reactivation, (again missing) can result in severe complications, especially CNS involvement.
L109: The serologic test of our patient showed a positive-IgG result and a grayzone result of 109 IgM.
What do you mean by grayzone? Please specify.
L111: available IgM detection technique, (comma missing) or the early examination
L.122: keep it in mind, (comma missing) when approaching patients
Author Response
Dear Reviewer, thanks for your comment and question. Please see the attachment for the correction of English language. The reply of your common and question will be presented here.Thank you.
Reviewer 1
- In 3% of children developing HZ a malignancy as underlying disease is discovered(especially leukemia). This might be worthwhile mentioning in the discussion, is research done in the case study?
- What research was done to detect concurrent immunsuppression?
- There have been several case studies who reported post covid HZ and even HZ as indicator of latent covid 19 infection(Elsaie et al. Dermatologic therapy/vol33/issu4). Is a covid test performed?
Reply for question 1 & 2
Dear Reviewer, thanks for your question. The patient was previously healthy with normal development curve, without any signs of immunodeficiency. He and his representative denied to have any medication, even Chinese herbs. We had complete blood count with differentiation, which displayed no young cells or blasts noted. Also, there were no signs which would suggest malignancy from history or physical examinations (eg. fever of unkown origin, bone pain, hepatosplenomegaly, petechiae…)
Reply for question 3
Dear Reviewer, thanks for your question. Indeed, asymptomatic infection cannot be ruled out (around 14.9% of pediatric population in a review article). But our patient denied to have traveled history, contact history of confirmed or suspicious cases, or any respiratory symptoms. His family also denied aforementioned condition. He also denied having other symptoms as abnormal or loss of smell/taste, diarrhea...which were recently suggested in consider of COVID-19 infection. Also, the diagnosed pediatric case in Taiwan were all confirmed to have contact or travel histories. Thus, we didn’t consider of COVID-19 infections.

Reviewer 2 Report
an interesting article, especially the review. what do you think is the possible mechanism of action by which these herpes zoster symptoms occur after the vaccine? is there any hypothesis?
Author Response
Dear Reviewer, thanks for your question. Since the vaccine was live-attenuated, it is thought to have a potential of latent infection. Although there was no clear mechanism demonstrated by previous research; we considered that these symptoms occurred similar as reactivation of previous varicella infection.
Sincerely,
Bing-Shiau Shang (1st author)
Cheng-Jui Jamie Hung (2nd author)
Ko-Huang Lue (Corresponding author)
Reviewer 3 Report
It is an interesting case report for the pediatric and dermatology clinic, but also for the clinical immunologist. The report would benefit from some paraclinical data of the patient, haematological count, C reactive protein levels, etc. The authors have not analyzed the possibility of a minor immunodeficiency and should discuss it. There are several reports on minor immune deficiencies that would affect the response of the patient that must be included. For example Körholz J, Richter N, Schäfer J, Schuetz C, Roesler J. A case of recurrent herpes simplex 2 encephalitis, VZV reactivations, and dominant partial interferon-gamma-receptor-1 deficiency supports relevance of IFNgamma for antiviral defense in humans. Mol Cell Pediatr. 2020 Oct 14;7(1):14.
Mogensen TH. IRF and STAT Transcription Factors - From Basic Biology to Roles in Infection, Protective Immunity, and Primary Immunodeficiencies. Front Immunol. 2019 Jan 8;9:3047.
Minor issues: The abstract should be modified since it is not clear. Minor details of the English language should also be corrected.
Author Response
Dear Reviewer, thanks for your professional opinion. We had noted that some minor immunodeficiency involved the pathway of innate immunity had susceptibility to specific pathogens, and the involved patient may presented previously healthy. Since most of the patient in such reports had recurrent or severe condition compared to our case, we did not consider this probability due to his first episode. Further, Droman et al. demonstrated that patient with IFNγR1 deficiency had over 70% of infection from environmental or BCG mycobacteria in both dominant and recessive type, with 100% of BCG vaccine-related mycobacteria disease in recessive inherited type. In Taiwan, we had BCG vaccination at 5-month-old infant scheduled in routine childhood vaccination program, and our patient denied any complications. However, thank you for reminding us the probability of certain immunodeficiency, and we may added this issue in our discussion. Thank you very much.
For minor issues, we will modify the abstract and the English language will be corrected as other reviewers also suggested. Thank you very much.
Dorman et al. Clinical features of dominant and recessive interferon gamma receptor 1 deficiencies. Lancet. 2004 Dec 11-17;364(9451):2113-21.
Sincerely,
Bing-Shiau Shang (1st author)
Cheng-Jui Jamie Hung (2nd author)
Ko-Huang Lue (Corresponding author)